# The Effect of Elevation on the Population Structure, Spatial Patterning and Intraspecific Interactions of *Picea schrenkiana* in the Eastern Tianshan Mountains: A Test of the Stress Gradient Hypothesis

Jianing He [1], Caiwen Ning [2], Wentao Zhang [1], Ümüt Halik [1] and Zehao Shen [2,*]

1 Key Laboratory of Oasis Ecology, Ministry of Education, College of Ecology and Environment, Xinjiang University, Urumchi 830000, China; hjn09325585560@163.com (J.H.); 18630005406@163.com (W.Z.); halik@xju.edu.cn (Ü.H.)
2 Key Laboratory for Earth Surface Processes, Ministry of Education, College of Urban and Environmental Sciences, Peking University, Beijing 100871, China; cwning@stu.pku.edu.cn
* Correspondence: shzh@urban.pku.edu.cn

**Abstract:** Changes in age structure, spatial distribution and intraspecific interactions across environmental gradients often reflect adaptations of plant populations to their environment. Our study explored the growth status of the *Picea schrenkiana* population on the north slope of the eastern Tianshan Mountains and tested the stress gradient hypothesis (SGH) against changes in the age structure and spatial pattern of *P. schrenkiana* populations along the environmental gradient. We sampled the forests at eight elevational locations, comprising a total of 24 plots of 30 × 30 m area from 1800 to 2500 m a.s.l. in the Jiangbulake region. By scanning the 3D structure of the forests and sampling tree rings in each plot, we precisely determined the spatial location and diameter of the breast height (DBH) of each *P. schrenkiana* individual. By fitting the DBH-age power model and g(r) function of the point pattern, we examined the age structure, spatial patterning and intraspecific interactions of local *P. schrenkiana* populations within each plot and their correlation with habitat parameters. The results indicate that (1) juveniles dominate the overall population density, age structure and spatial patterning of the *P. schrenkiana* population. Trees of low–middle elevations represent younger forests with faster growth and better regeneration, while trees at high elevations form older forests with slower growth and poorer regeneration. (2) The aggregated population patterns and positive intraspecific interactions occur mostly at medium elevations (2000 and 2100 m a.s.l.). (3) Population density, aggregation intensity and intraspecific interaction strength are strongly and positively correlated ($p < 0.01$). Our results did not fit the SGH but support a hump-shaped hypothesis that proposes that facilitation is stronger under medium stress along the elevational gradient. This study validates the spatial point pattern testing of the SGH of different types. We recommend the implementation of more intensive forest closure measures, together with a reduction in the harvesting intensity of trees to ensure the sustainable regeneration of *P. schrenkiana* forests in the eastern Tianshan Mountains.

**Keywords:** elevation; intraspecific interaction; regeneration; Tianshan Mountains; growth

## 1. Introduction

The study of a population size, distribution and interactions with biotic and abiotic factors falls under the purview of population ecology [1]. Understanding the survival status and environmental requirements of a species requires a comprehensive knowledge of its population structure and spatial patterns [2–4]. Age structure, which refers to the distribution of different age groups within a population, serves as a crucial indicator for characterizing future population trends [5]. A population's spatial distribution ultimately

results from the interactions of its individuals with their immediate environment [6–8]. Due to competition among nearby plants for light, soil nutrients, and water resources, plants respond immediately to their close neighbors [8,9]. The theory of population density dependence has long explained phenomena like competition and facilitation [10]. Competition is the primary driving force determining species cohabitation and peer group self-thinning [11]. In contrast, facilitation can mitigate the negative effects of environmental stress and resource competition [12]. An example of this is the occupation of habitats by highly light-tolerant plants, whose canopy can provide a more ideal light environment for the growth and regeneration of shade-tolerant plants of lower stature [13].

The degree of abiotic stress experienced by interacting individuals significantly affects the balance between facilitation and competition [14–16]. According to the stress gradient hypothesis (SGH), facilitation prevails when the ambient stress is strong, whereas competition dominates in low-stress environments [10,16]. Although the majority of studies support the SGH [17–19], some research supports a hump-shaped hypothesis that proposes that facilitation is stronger under medium stress [16,18,20,21], and facilitation will break down when environmental forces surpass a specific threshold [17,20,22]. The refined SGH indicates that facilitation under environmental stress may be influenced by the plant's life strategy or the type of stress, rather than the actual degree of stress [23,24]. Notably, some research demonstrated through point pattern analyses that species associations in harsher environments align with some predictions of the SGH [25]. In other research, both competitive and facilitative interactions were explored within the same or different species [11,19,26–28]. For instance, some research observed variations in spatial associations between shrub species and pine seedlings along an aridity gradient that was consistent with the SGH [19]. There are also studies that found facilitation was more prevalent than competition among shade-tolerant species in the boreal forests in northern Xinjiang of Northwest China [27].

Elevational gradients that span a species' niche offer an excellent opportunity to examine changes in the spatial associations of populations because they can naturally produce variations in habitat conditions from core suitability zones to marginal critical habitats, which are reflected in the variations in the population structure and patterns [29]. Such conditions provide natural temperature gradient stress, and therefore, the possibility of examining the consistency of the changes in intraspecific spatial positive and negative effects with the SGH. Moreover, the environmental factors which dominate species' ecological niche changes across altitudinal gradients could be determined. Currently, the number of relevant studies on this topic is limited.

The subalpine forest belt of the Tianshan Mountains is mainly dominated by *Picea schrenkiana* Fischet Mey, an indigenous tree species of the region. Additionally, *P. schrenkiana* plays a vital role in the forest ecosystems of the Tianshan Mountains and serves as the primary biological carbon bank in Xinjiang, China, contributing to biodiversity maintenance, regional water conservation and climate regulation [30]. This study aims to investigate age structure and spatial patterning in the *P. schrenkiana* population through a sample survey of pure forests on the elevational gradient of the northeast slope of the eastern Tianshan Mountains, while also testing the relationship between the spatial correlation and SGH. Based on prior research, we hypothesize that (1) the regeneration status and age structure of populations significantly vary along the elevational gradient, with faster tree growth at lower elevations. (2) The spatial association of populations along the elevational gradient might support the hump-shaped hypothesis about stress gradients. (3) The strength of spatial interactions between trees of different diameter of breast height (DBH) classes may significantly correlate with the degree of spatial aggregation in populations.

## 2. Materials and Methods

### 2.1. Study Area

The study area was located in Jiangbulake National Forest Park (43°25′06″–43°39′42″ N, 89°25′15″–90°16′49″ E) at the northern foot of the eastern Tianshan Mountains in Xinjiang,

China, with an area of about 4800 hm2. The annual mean temperature in this area is about 0–3 °C, and the annual precipitation is about 250–500 mm. The vertical spectrum of vegetation belts on the northern slope of the Tianshan Mountain is defined as temperate desert (700–1100 m a.s.l.), montane grassland (1100–1650 m a.s.l.), montane coniferous forest (1650–2700 m a.s.l.) and subalpine meadow (2700–2900 m a.s.l.). The montane coniferous forest belt distributed at 1650–2700 m is mainly a single-established species forest composed of *P. schrenkiana*, which is a shade-tolerant species endemic to the Tianshan Mountains.

From July to August 2022, a total of 24 sample plots were established in the forested area (1800–2500 m a.s.l.), with three repeating sample plots at each of eight elevations, separated by an elevation interval of 100 m. The size of each sample plot was 30 × 30 m with a total area of 21,600 m$^2$. At 2600 m a.s.l, there were too few trees for tree point analysis due to the proximity of the alpine tree line limit but tree core sampling was conducted.

About 25 tree cores of different ages were drilled with growth cones in each sample plot. The 3D structure of the forest community in the sample plots was then scanned and measured using a LiBackpack (GreenValley International, Berkeley, CA, USA), and the point cloud data were de-noised, cropped and normalized using Lidar360 software v.6 (GreenValley, Beijing, China) before being split into single trees [31]. The location, DBH, tree height and crown spread of each tree (DBH $\geq$ 5 cm) were obtained. They were classified into 8 DBH classes and age classes according to previous studies [32]: DBH class: $5 < I \leq 10$ cm, $10 < II \leq 15$, $15 < III \leq 20$ cm, $20 < IV \leq 25$ cm, $25 < V \leq 30$ cm, $30 < VI \leq 35$ cm, $35 < VII \leq 40$ cm, $40$ cm $< VIII$; age class (yrs): A (0–20), B (20–40), C (40–60), D (60–80), E (80–100), F (100–120), G (120–140), H (>140). For the analysis of spatial patterns and intraspecific correlations, *P. schrenkiana* trees in the sample plots were classified into two DBH groups of juveniles ($5 \leq DBH \leq 15$ cm) and adults (DBH > 15 cm).

### 2.2. Methods

#### 2.2.1. Point Pattern Analysis and Null Model (Completely Spatial Randomness, CSR)

In order to understand the characteristics of small-scale patterns, the g(r) function was used [33]. The g(r) function is a second-order statistic closely related to the first derivative of Ripley's K(r) function [34]. It improves the cumulative effect of the K(r) function at the spatial scale by taking the distribution of all individuals within a circle with radius r and width dw. We used the univariate $g_1(r)$ function to analyze the spatial distribution pattern of the population at the sample site scale and the pairwise correlation $g_{12}(r)$ function to analyze the spatial correlation between juveniles and adults of the population; the completely random model assumes that the probability of the occurrence of any point in the mid-space is the same, and our sample sites were in forested areas with relatively homogeneous environments, so the completely random model was used as a descriptor of the degree of data bias in the null model [35]. We classified the spatial pattern and correlation scales into three scales: small scale (S, $0 < r \leq 2$ m), medium scale (M, $2 < r \leq 6$ m) and large scale (L, $r > 6$ m). Spatial point pattern and correlation analyses were conducted in R4.2.2 (R Foundation for Statistical Computing, Vienna, Austria).

$$K_1(r) = \frac{A}{n^2} \sum_{i=1}^{n} \sum_{j=1}^{n} \frac{1}{W_{ij}} I_r(u_{ij})(i \neq j) \tag{1}$$

$$K_{12}(r) = \frac{A}{n_1 n_2} \sum_{i=1}^{n} \sum_{j=1}^{n} \frac{1}{W_{ij}} I_r(u_{ij})(i \neq j) \tag{2}$$

where A is the area of the plot (see Equations (1) and (2)); n is the total number of individuals in the plot; $n_1$ and $n_2$ are the numbers of trees with different DBHs; uij is the distance between two random points; and Ir (u$_{ij}$) is an indicator function. If $u_{ij} \leq r$, Ir (uij) = 1, and if $u_{ij} > r$, Ir (u$_{ij}$) = 0; W$_{ij}$ is the adjustment for boundary effects.

$$g_1(r) = \frac{1}{2\pi r} \times \frac{dK_1(r)}{dr} \tag{3}$$

$$g_{12}(r) = \frac{1}{2\pi r} \times \frac{dK_{12}(r)}{dr} \qquad (4)$$

where $dK_1(r)$ and $dK_{12}(r)$ are the derivatives of $K_1(r)$ and $K_{12}(r)$ (see Equations (3) and (4)), respectively. If $g(r) > 1$, individuals follow a clustered distribution; if $g(r) = 1$, a random distribution; if $g(r) < 1$, individuals follow a uniform distribution. The function was used in bivariate scatterplot analysis. To evaluate the significance level of the analysis statistics under the null model, 99% confidence intervals were obtained after 199 Monte-Carlo simulations. A number above the confidence interval indicates a positive correlation, while a number below the confidence interval indicates a negative correlation. Points within the confidence interval indicate no significant correlation [36].

### 2.2.2. Growth Rate and Correlation Analysis

The cores drilled from the growth cones were fixed with wood glue in wooden grooves and sequentially sanded with sandpaper of 200, 600 and 1200 mesh until the annual rings were visible, so that the tree age could be read under a $10\times$ magnifying glass.

There is a positively correlated relationship between age and DBH [37]; DBH-age single-wood growth models of *P. schrenkiana* populations at different elevations were fitted with power indices to estimate the age of the trees [32]. The growth of *P. schrenkiana* populations at each elevation was estimated from the average growth rate (the average ratio of DBH to the age of trees drilled at each elevation) (Table 1).

$$SA = \frac{\sum_0^r g_1(r)}{r} \qquad (5)$$

$$SI = \frac{\sum_0^r g_{12}(r)}{r} \qquad (6)$$

**Table 1.** Power models of DBH and age relationship and radial growth rate of *P. schrenkiana* at different elevations.

| Elevation (m) | Model | $R^2$ | F | $p$ | Average Growth Rate (cm/a) |
|---|---|---|---|---|---|
| 1800 | $D = 0.473\ A^{0.952}$ | 0.364 | 36.108 | 0.000 | 0.4047 |
| 1900 | $D = 2.411\ A^{0.587}$ | 0.341 | 29.989 | 0.000 | 0.4417 |
| 2000 | $D = 0.797\ A^{0.842}$ | 0.525 | 73.978 | 0.000 | 0.4333 |
| 2100 | $D = 0.392\ A^{1.05}$ | 0.689 | 137.269 | 0.000 | 0.4973 |
| 2200 | $D = 0.328\ A^{0.951}$ | 0.385 | 31.277 | 0.000 | 0.2707 |
| 2300 | $D = 0.154\ A^{1.082}$ | 0.513 | 59.996 | 0.000 | 0.2431 |
| 2400 | $D = 10.293\ A^{0.159}$ | 0.074 | 5.037 | 0.028 | 0.1707 |
| 2500 | $D = 0.614\ A^{0.756}$ | 0.563 | 81 | 0.000 | 0.2004 |
| 2600 | $D = 0.961\ A^{0.795}$ | 0.738 | 87.29 | 0.000 | 0.3991 |

The strengths of tree aggregation (SA) and interaction (SI) between the two DBH classes at each sample site were characterized from the means of the $g_1(r)$ and $g_{12}(r)$ functions and obtained from sites at 1 m intervals (see Equations (5) and (6)) [25]. The average nearest neighbor distance (ANND) was calculated using the "dist" function in R4.2.2 to characterize the strength of the clustering of juveniles (J_ANND) and adults (A_ANND) in the sample plot. For correlation analysis, we used J/A_N for the number of juveniles or adults, F_N for the number of tree falls, PD for population density, J_DNN for the number of natural juvenile deaths, and M_DBH for the mean DBH.

## 3. Results

### 3.1. Population Age Structure and Growth

Overall, the population density of *P. schrenkiana* varied significantly with elevation (Figure 1). There was a clear pattern of population aggregation of larger trees at 2100 m a.s.l., whereas the population densities were obviously lower at an elevation of 1900 and 2200 m. Trees were more evenly distributed at an elevation of 2400 m, and there were obviously fewer trees of large DBH classes at 2500 m.

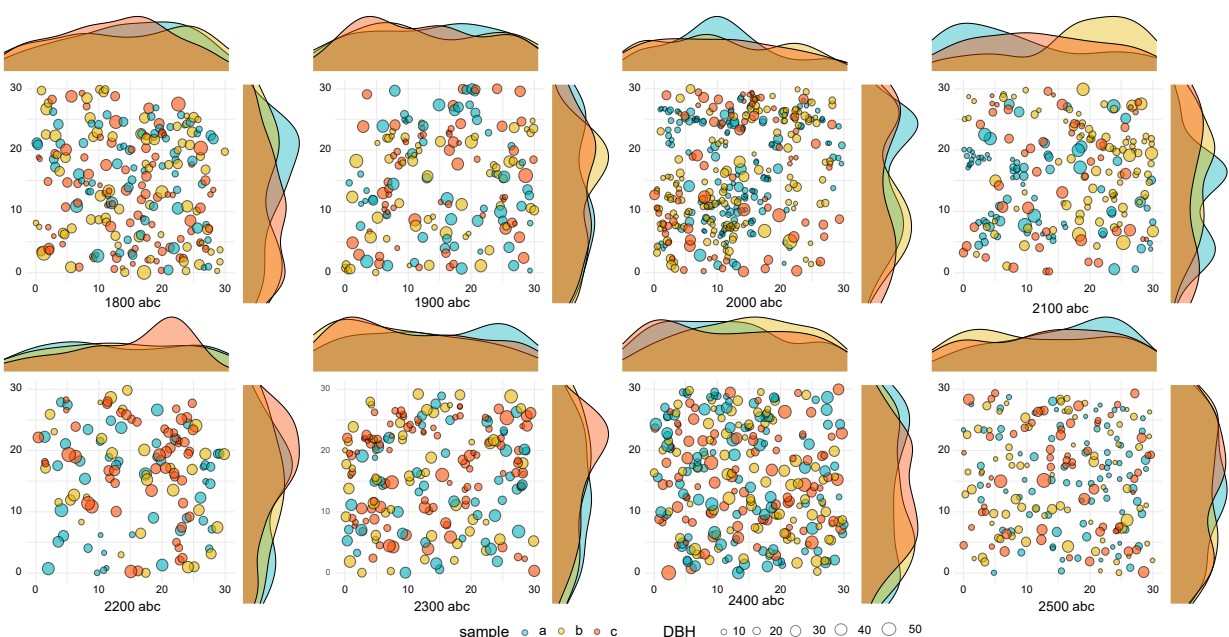

**Figure 1.** Population distribution map of all 24 plots at an elevation of 1800–2500 m. The different colors represent data from three plots at the same elevation (plots a, b and c). Circle sizes are scaled according to DBH. The density curves along both sides of the map represent the density distribution of the population distribution in the three (30 × 30 m) plots. The number below each panel represents the elevation of the plot.

To examine differences in age structure (Figure 2), we divided the *P. schrenkiana* population into two spatial zones: low and middle elevation (1800–2100 m) and high elevation (2200–2500 m). Overall, the number of A-age trees at low–middle elevations was much higher than that at high elevations, with fewer old trees, indicating that the population at low–middle elevations displayed better regeneration and the local forest was younger. The number of E-H-age trees at higher elevations was significantly higher than that at low and middle elevations and consisted mainly of older forest. However, trees at the higher elevations tended to achieve smaller DBH with the same age, indicating an increasing restriction of radial growth.

The power function model fitted the DBH-age relationship of the *P. schrenkiana* population well at all elevations (Table 1) except at elevation 2400 (F, *p* values), suggesting that the variation in tree DBH of the same age class at this elevation was relatively large. We calculated the average annual radial growth rate of the local *P. schrenkiana* population at each elevation (Table 1), and overall, the radial growth rate of trees at middle and low elevations was higher than that at high elevations. Tree growth was fastest at 2100 m above sea level and slowest at 2400 m above sea level, while the average growth rates at the upper and lower tree lines (1800 and 2600 m) were similar.

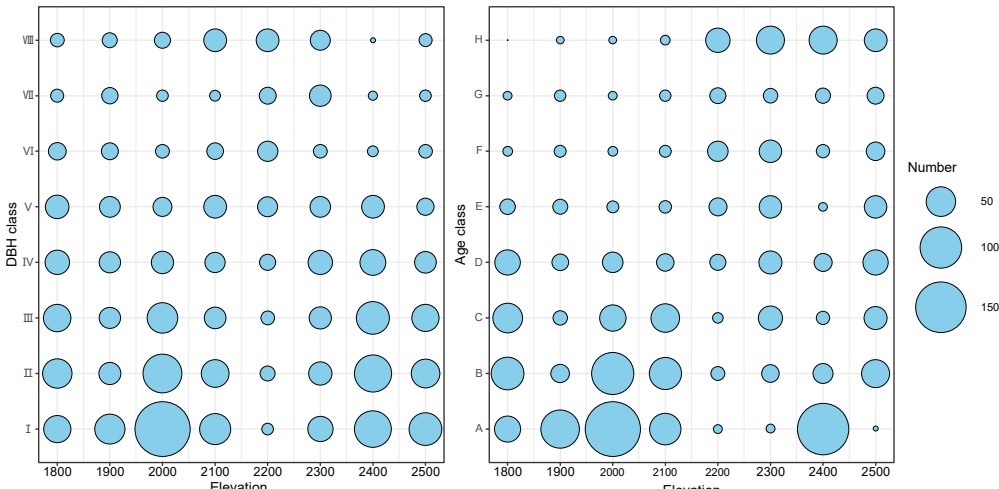

**Figure 2.** The distribution of DBH class and age class of *P. schrenkiana* at an elevation of 1800–2500 m. The size of the blue circle represents the number of trees in a particular DBH or age class at that elevation.

### 3.2. Spatial Patterns and Intraspecific Interaction

The spatial patterns of *P. scherenkiana* varied across the elevational gradient; the clustered pattern was concentrated at the elevations of 2000 and 2100 m, while a random distribution pattern dominated at both lower and higher elevations (Figure 3).

According to the scales defined for the spatial patterns of the populations, juveniles mainly occurred in clustered patterns at small and medium scales (S: 0 < r ≤ 2 m and M: 2 < r ≤ 6 m), and mainly at low and middle elevations, whereas at the larger scale (L: r > 6 m) and at higher elevations, the juveniles displayed a random distribution. In contrast, adults almost always displayed a random distribution pattern at all scales and elevations (Figure 3, Table 2).

**Table 2.** Spatial distribution pattern of juveniles and adults.

| | Juveniles | | | Adults | | | | Juveniles | | | Adults | | |
|---|---|---|---|---|---|---|---|---|---|---|---|---|---|
| Scale | S | M | L | S | M | L | Plot | S | M | L | S | M | L |
| 1800a | C | C | R | R | R | R | 2200a | — | — | — | R | R | R |
| 1800b | C | C | R | R | R | R | 2200b | — | — | — | R | R | R |
| 1800c | R | R | R | R | R | R | 2200c | — | — | — | C | R | R |
| 1900a | R | R | R | R | R | R | 2300a | — | — | — | R | R | R |
| 1900b | C | R | R | C | R | R | 2300b | — | — | — | R | R | R |
| 1900c | C | R | R | R | R | R | 2300c | C | C | C | R | R | R |
| 2000a | C | C | R | R | R | R | 2400a | C | R | R | R | R | R |
| 2000b | C | C | R | C | R | R | 2400b | R | C | R | R | R | R |
| 2000c | C | C | R | R | R | R | 2400c | R | R | R | R | C | R |
| 2100a | C | C | C | R | R | R | 2500a | R | C | R | R | R | R |
| 2100b | C | C | C | R | R | C | 2500b | R | R | R | R | R | R |
| 2100c | — | — | — | R | R | R | 2500c | R | R | R | R | R | R |

Note: C: clumped; R: random; —: the number is too small to use point pattern analysis; a,b,c: plot a,b and c at the same altitude.

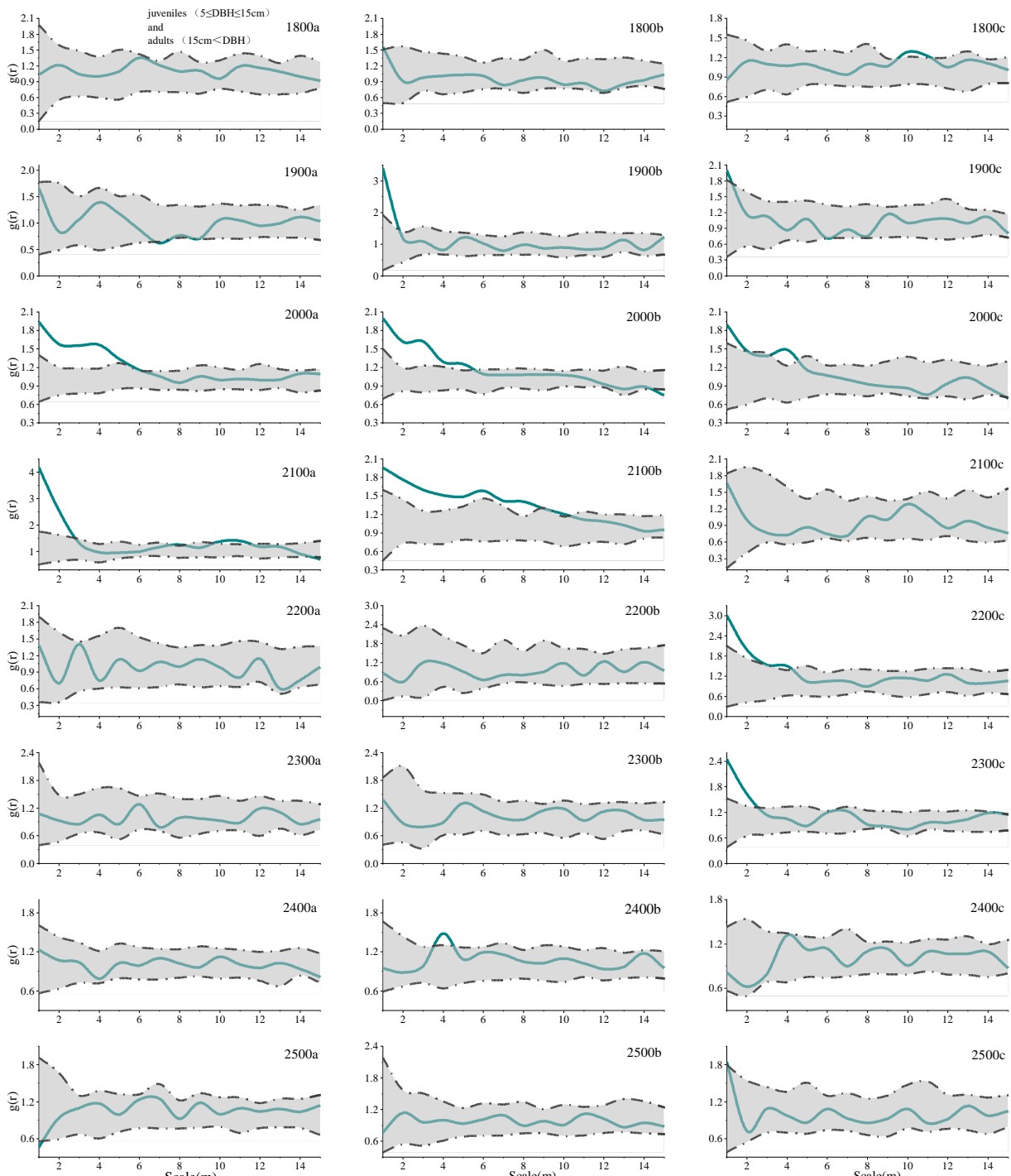

**Figure 3.** Spatial pattern analysis of each plot at different elevations based on g(r) functions. a,b,c: plot a,b and c at the same altitude.

Negative associations between juveniles and adults rarely occurred. Positive associations were infrequently observed at low (1800–1900 m) and high (2200–2500 m) elevations but occurred mainly at the elevations of 2000 and 2100 m (Figure 4). This positive association is consistent with the higher frequency of clustering at these elevations, suggesting that positive intraspecific interactions firstly increase with elevation and then diminish (Figure 4).

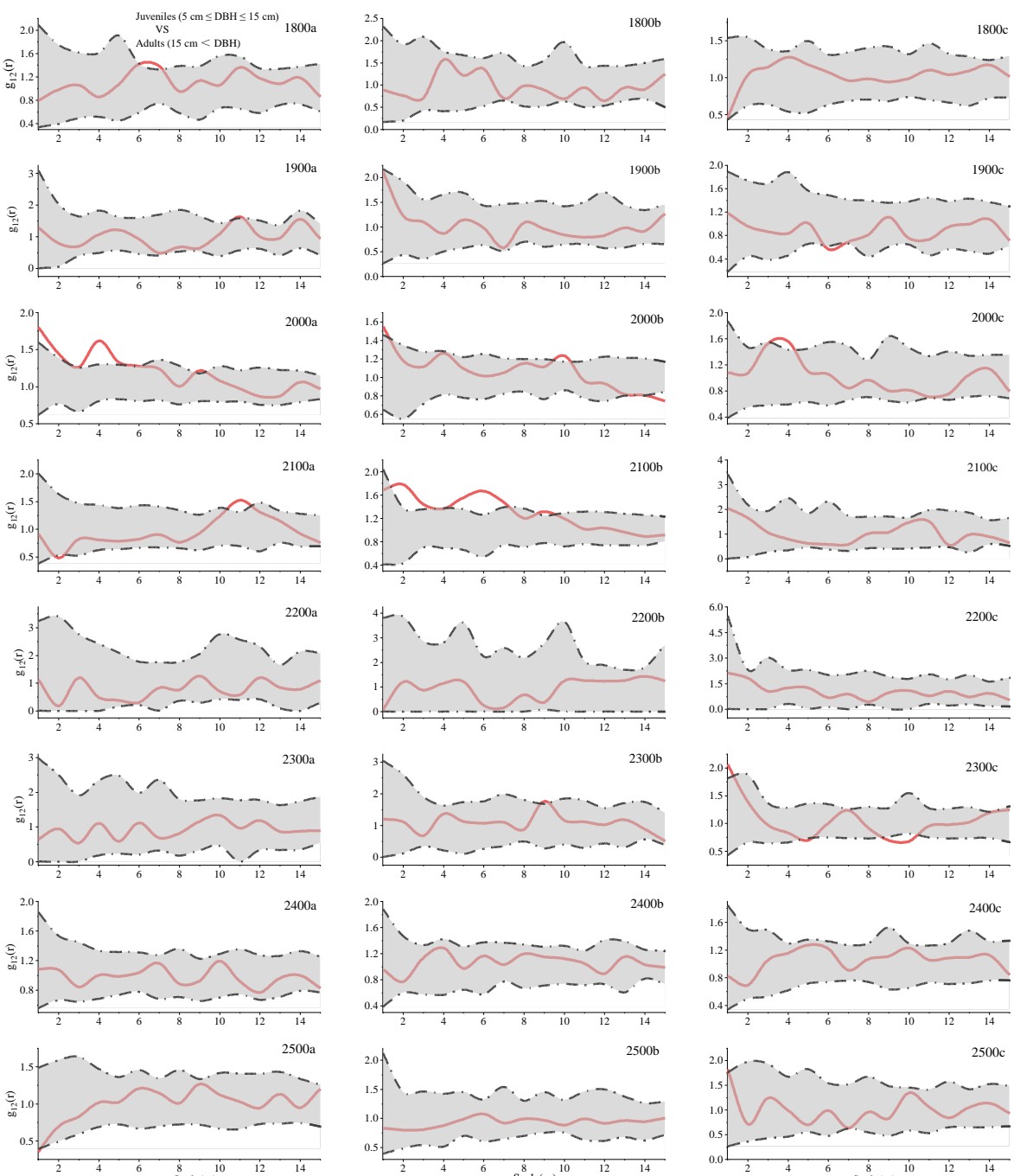

**Figure 4.** Intraspecific interactions of each plot at different elevations among two size classes. a,b,c: plot a,b and c at the same altitude.

### 3.3. Correlation Analysis of Different Habitat Factors

The Pearson correlation coefficients between the indicators of each sample site were found (Figure 5). The SA displayed highly significant correlations with SI (strength of the interaction) and J_ANND and a significantly correlation with A_ANND (adults average nearest neighbor distance), indicating that SA (strength of aggregation) was closely related to SI. The aggregation patterns across sites reflect the joint contributions of juveniles and adults, but with juveniles making a greater and more significant contribution. SI was significantly correlated with J_N (juvenile number), the number of J_DNN (juvenile natural deaths), the PD (population density), and the mean DBH, indicating that the SI was regulated by several factors. However, it appeared to have significant correlations

with J_DNN and PD (positive), J_ANND (juveniles' average nearest neighbor distance) and M_DBH (negative). J_DNN showed a significantly positive correlation with the PD and J_ANND, showing that the aggregation pattern was the result of the joint action of juveniles and adults within the sample site. The J_DNN was significantly correlated with the PD and displayed a significant negative correlation with the J_ANND, which indicated that density dependence led to competitive exclusion and the deaths of juveniles. The correlation coefficients between A_N, FN (number of felled trees) and other factors were small, indicating that the number of cut trees that fell in the sample plots did not significantly promote the regeneration of juveniles.

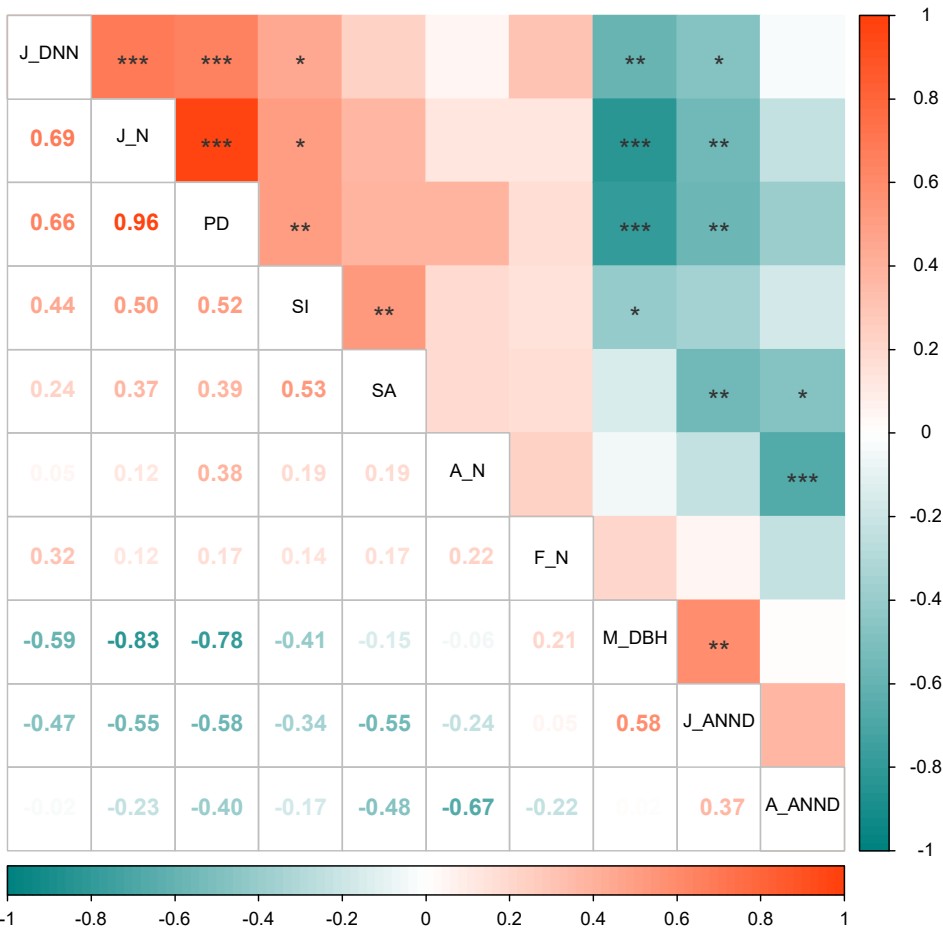

**Figure 5.** The correlations between the population pattern parameters of the plots. The correlations are given to the lower left and their significance to the upper right, where ***, ** and * correspond to *p* < 0.001, *p* < 0.01, and *p* < 0.05, respectively. Plot indicator key: J/A_N: juvenile or adult number; F_N: number of felled trees; PD: population density; J_DNN: number of natural juvenile deaths; SA: strength of aggregation; SI: strength of the interaction; J/A_ANND: juveniles' or adults' average nearest neighbor distance; M_DBH: mean DBH.

## 4. Discussion

### 4.1. Population Age Structure and Growth

Except for the elevation of 2000 m a.s.l., there were more small DBH trees at each elevation, which had younger ages in the low and middle elevations and older ages in the higher elevations. The mean radial growth rate of trees at low and middle elevations was significantly higher than that at higher elevations, leading to distinct differences in the age structure along the elevation gradient. In the central Tianshan Mountains, *P. schrenkiana* is distributed at elevations between 1500 and 1700 m [32]. However, in our study area, tree growth was limited to approximately 1800 m above sea level, indicating a narrower *P.*

*schrenkiana* population distribution across elevations in the eastern Tianshan Mountains. Notably, a large number of trees were found at 2400 m, but with significant variation in the age of trees with the same DBH class, meaning the *P. schrenkiana* growth model showed the poorest fit (F, p), resulting in the lower accuracy of tree age prediction at this elevation. As a result, there may not have been as many young trees (age class A) as the age structure suggests (Figure 2).

The DBH class structures of all elevation segments, except for 2200 and 2300 m, indicated growing populations, which contradicts the results indicated by the age structure. The use of DBH class, instead of age class, did not effectively characterize the regeneration status of the populations. Differences in age and class structure are often attributed to variations in the radial growth rates of trees in different habitats [38]. Studies have shown that changes in temperature, light, precipitation, soil nutrients and other factors related to elevational gradients generally result in slower growth rates of tree DBH at higher elevations compared to middle and lower elevations [39,40]. Consequently, trees at higher elevations tend to have smaller DBHs but older ages.

The average radial growth rate of the *P. schrenkiana* population at elevations of 2100–2500 m exhibited a rapid decrease, followed by an increase and subsequent decrease. The initial decline in radial growth rate is likely attributed to drought conditions caused by climate warming, which hinders tree growth [41]. Conversely, the increased precipitation at higher elevations could support higher tree growth rates. However, at an elevation of 2400 m and above, lower temperatures began to hinder the radial growth of *P. schrenkiana* trees. Surprisingly, the sparsely wooded upper treeline (at the elevation 2600 m), which experiences the lowest temperatures within the distribution range of *P. schrenkiana*, supported a greater radial growth rate compared to elevations of 2200–2500 m, indicating that the lower temperatures at the upper treeline promoted rather than inhibited the radial growth of *P. schrenkiana*. A study also found that climate warming was a significant driver of accelerated growth in the upper treeline of the Tianshan Mountains, where the lower population density and reduced competition may additionally favor radial growth [42]. However, the population of *P. schrenkiana* still displayed faster growth in the middle and lower elevations. The radial growth of trees is influenced by various environmental factors, including temperature, precipitation, and soil properties, with different factors dominating at different elevations. Nonetheless, the specific ecological factors and underlying mechanisms governing the radial growth rate of *P. schrenkiana* populations at various elevations require further investigation.

### 4.2. Spatial Distribution Pattern

The overall pattern of *P. schrenkiana* populations is largely influenced by sample size effects, wherein the clustering pattern at small to medium scales is dominated by juveniles [43,44]. The spatial distribution of juveniles is primarily influenced by seed dispersal modes and heterogeneous environments [45]. Seeds of spruces are more likely to be dispersed from cones after shedding, explaining the clustering of juveniles in the *P. schrenkiana* population. Within the sample plots, small trees tend to cluster near larger trees rather than in more spacious plots (Figure 1, 2100a), suggesting the formation of suitable habitats for young trees to concentrate and thrive under the shelter of adult crowns. The middle and lower elevations provide a more favorable environment for seed germination, resulting in a higher abundance of juveniles and faster radial growth. Thus, the portion of the population between 1800 and 2100 m is in an optimal ecological location. Conversely, spatial aggregation occurs less frequently at high elevation, which is likely due to mature old-growth forests and the early completion of population self-thinning through competition [46,47]. Additionally, the harsh habitat at high elevations may limit the growth and survival of *P. schrenkiana* seedlings [32].

The spatial distribution patterns of mature trees are influenced by both biotic and abiotic factors, including environmental attributes [8,9,48]. The even distribution of adults can be a result of the self-thinning mortality of populations, as well as the presence of

relatively homogeneous environments. In highly heterogeneous environments, adults may aggregate in plots with better soil nutrients. For instance, there is a clear pattern of aggregated distribution of adult *Populus euphratica* in fragmented habitats [7]. In the case of *P. schrenkiana*, the spatial clustering of adults is observed relatively infrequently, indicating that density-dependent self-thinning is the primary reason for the even distribution of adults.

### 4.3. Intraspecific Interactions and SGH

Altitude is a natural temperature gradient. We observed that spatial positive associations of the local *P. schrenkiana* population tend to initially increase and then decrease with elevation. This finding rejects the SGH but aligns with the hump-type stress hypothesis prediction [16,22], which suggests that plant facilitation is more pronounced in moderately stressful (temperature) environments. These results indicate that, to some extent, the local spatial patterning of the forest reflects the environmental stress, which is consistent with a previous study [25]. However, the spatial correlation analyses might only indicate a possible relationship, and a direct test of the stress gradient hypothesis requires further mechanistic exploration.

Positive effects play a crucial role in plant adaptation to prolonged cold and dry climates [26,28,30]. This kind of intraspecific facilitation is less common in tropical and subtropical forests but is more likely to occur in harsh environments [49,50], particularly between individuals of different sizes [27,51,52]. Positive effects between large and small individuals in pure stands of *P. schrenkiana* may represent an adaptive survival mode for the species in its extended winter environment [27]. However, this mechanism exhibited significant variation across the elevational gradient.

In this study, the intensity of intraspecific interactions was significantly correlated with population density, aggregation intensity, the number of natural juvenile deaths, the number of juveniles, and the mean DBH. These findings indicate that biotic factors regulate the intensity of intraspecific interactions. The changes in intraspecific interactions with elevation might not be caused solely by a single environmental stress but rather by the plant's life strategy as well. Some researchers attempted to reconcile the role of environmental stress versus life strategies in intraspecific interactions; this effort resulted in further controversy [17,53]. Hence, the mechanism of the competition–facilitation relationship appears to be complex, involving multiple factors.

In general, seedling density and survival tend to increase with the intensity of forest harvesting [54]. However, in the sample plots of this study, the number of logged trees was not significantly correlated with the number of juveniles. It is almost common knowledge that the *P. schrenkiana* is a shade-tolerant species. The regeneration of the *P. schrenkiana* population may be influenced by two factors. First, the growth characteristics of the trees play a role, as seedlings of shade-tolerant species depend on the shading provided by large trees. Second, the regeneration of *P. schrenkiana* individuals may be constrained by the altitude of their habitat. The regeneration of shade-tolerant tree species is influenced by various factors [55,56]. Therefore, we believe that the regeneration of *P. schrenkiana* populations is dependent on the shade toleration characteristics of the trees and the habitat suitability, rather than being solely influenced by the logging-induced light and growing space.

The *P. schrenkiana* forest in Jiangbulake exhibits structural homogeneity and faces challenges of extensive anthropogenic activities and high grazing intensity. We propose the following recommendations for the late-season management of the forest park: (1) Implement more intensive forest closures to reduce human and livestock-induced forest land destruction. (2) The felling strategy should consider the light sensitivity of the juveniles to ensure the maintenance of an adequate local adult canopy density.

## 5. Conclusions

　　In this study, we observed that *P. schrenkiana* populations exhibit faster growth and better regeneration statuses at low–middle elevations. Juveniles dominate the population density, age structure and spatial distribution, playing a crucial role in regulating the overall population structure and spatial patterning. Population competition is mainly evident through density-dependent mortality among juveniles. Notably, we found that spatial positive interactions of *P. schrenkiana* populations were more frequent at elevations of 2000 and 2100 m a.s.l., supporting the hypothesis of a hump-type pressure effect. Furthermore, population density and aggregation strength demonstrated a highly significant and positive correlation with the intensity of intraspecific interactions ($p < 0.01$), highlighting the influence of biotic factors on intraspecific interactions. Overall, our study provides additional evidence for point pattern testing the SGH and provides a theoretical basis for future sustainable forest management.

**Author Contributions:** J.H. and Z.S. conceived, designed and performed the experiments; collected and analyzed the data; prepared figures and tables; and wrote the manuscript draft. J.H., C.N., Ü.H. and W.Z. participated in field work and contributed to data collection, analysis, and the manuscript writing. Z.S. read the manuscript with critical comments and reviewed the manuscript draft. All authors have read and agreed to the published version of the manuscript.

**Funding:** This study was supported by the Second Comprehensive Scientific Expedition of the Qinghai–Tibet Plateau (2019QZKK04020101) and the National Natural Science Foundation of China under the major project topic (41790425).

**Data Availability Statement:** The data presented in this study are available upon reasonable request from the authors.

**Acknowledgments:** We thank the managers of Jiangbulake Scenic Area for their full support in data collection and Yuyang Xie for his guidance in processing the data in Lidar360 (GreenValley, Beijing, China).

**Conflicts of Interest:** The authors declare no conflict of interest.

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
