# Peer review of "The Effect of Elevation on the Population Structure, Spatial Patterning and Intraspecific Interactions of Picea schrenkiana in the Eastern Tianshan Mountains: A Test of the Stress Gradient Hypothesis"

_forests, doi:10.3390/f14102092_

Round 1
Reviewer 1 Report
This study provides valuable insights into the age structure, spatial distribution, and intraspecific interactions of P. schrenkiana populations in the eastern Tianshan Mountains. The findings suggest that younger forests with faster growth and better regeneration are predominant at lower elevations, while older forests with slower growth and poorer regeneration are found at higher elevations. The occurrence of positive intraspecific interactions at medium elevations highlights the importance of facilitation in these populations. The study recommends implementing more intensive forest closure measures and reducing harvesting intensity to ensure the sustainable regeneration of P. schrenkiana forests in the region.
Overall, while not entirely groundbreaking, this paper adds to our comprehension of plant population dynamics and enhances our understanding within the framework of the Stress Gradient Hypothesis.
My reccomendation is acceptance after minor revision.
Speciffic comments:
Lines 41-43 Please provide reference
Line 82 Please use italic for latin names of the plant species.
Author Response
Thank you for your valuable comments on the manuscript. Based on your comments, I have made revisions.
1) Lines 41-43 Please provide reference : I already finished providing.
2) Line 82 Please use italic for latin names of the plant species : I have finished.

Reviewer 2 Report
The manuscript (MS) may present interest for the readers of Forests dealing with mountain forest ecology, studies of spatial and age structure and altitudinal gradients in forest science. However, the MS requires some improvements before the publication.
1. The MS ignores a large literature of late 1980s and 1990s dedicated to the use of Ripley's function and its modifications to the analysis of spatial structure of various forest stands (for example the works of Prof. Dr. D. Stoyan and his students). In principle, it is not an obstacle for a publication, but it reduces the value of the research.
2. p.1, l.14, p.12, l. 336 -> it's a standard to provide the authors' names at the first appearance of the Latin species names.
3. p.2, l.61 -> indicates?
4. p.2, l.67,69, p.11, l.306, p.12, l.337,362 -> format of citing.
5. p.2, l.82: PiceaSchrenkiana.
6. p.2, l.89-93: we hypothesize that ... (2) The spatial association of populations along the elevational gradient might support certain hypotheses about stress gradients -> 'certain hypotheses' is a bit vague, it's better to specify which hypothesis.
7. p.3, l.103 -> mountain?
8. p.3, l.111 -> please, specify units 'hm2'.
9. p.3, l.144-148 -> the formulas (1) and (2) are completely the same.
10. p.4, l.151 -> incomplete sentence.
11. p.4, l.176: There is a strong quantitative relationship -> this is a bit too strong assertion. The relationships are really highly significant but R2 says that the model explains about a half or lower share of data variation.
12. p.4, l.181 -> it's better to give SA and SI as formulas because they are important pieces of the analysis.
13. p.4, l.183: functions at -> and?
14. p.4, l.188 -> if the trees were reconstructed from LIDAR data then how the dead trees were recognized?
15. p.5, l.206 -> regeneration.
16. p.5, Figure 1 -> it's rather inconvenient to consider the overlapped maps. Probably, it's better to show only one of them and others to place into a supplement.
17. p.6, l.228: P. scherenkiana
18. p.9-10, l.249-265 -> it's really hard to read the paragraph with so many abbreviations. The reader has to remember too many of them she pages earlier. Probably, placing the information into a table would be better.
19. p.11, l.310: lower elevations, The radial growth -> full stop instead of comma?
20. p.12, l.369: tolerantspecies
Overall, English is quite comprehensible. Minor editing of English language required.
Round 2
Reviewer 2 Report
I have read the answers by the authors and I believe that the manuscript has been improved. However, some small drawbacks are clear for correction but it hasn't be done.
For example, l. 111: 2.16 hm2 -> what are these units? It's a strange combination. 30 m X 30 m X 24 plots = 21600 m2 or 2.16 ha. Why do the authors insist on hm2?
Overall, the paper may be further technically edited and accepted.
English is well understandable.
Author Response
Thank you for your valuable comments on non-standard terms, and the author has revised it in time.
